# The Influence of Psychopathy on Incarcerated Inmates’ Cognitive Empathy

**DOI:** 10.3390/brainsci12081003

**Published:** 2022-07-28

**Authors:** Gerardo Flórez, Ventura Ferrer, Luis García, María Crespo, Manuel Pérez, Pilar Saiz

**Affiliations:** 1Centro de Investigación Biomédica en Red de Salud Mental (CIBERSAM), 33011 Oviedo, Spain; frank@uniovi.es; 2Health Department, Pereiro de Aguiar Prison, 32792 Ourense, Spain; ventura.ferrer@gmail.com (V.F.); medico.orense@dgip.mir.es (L.G.); maria.crespo.leiro@gmail.com (M.C.); maperi1@msn.com (M.P.); 3Department of Psychiatry, University of Oviedo, 33003 Oviedo, Spain; 4Instituto de Investigación Sanitaria del Principado de Asturias (ISPA), 33011 Oviedo, Spain; 5Mental Health Services of Principado de Asturias (SESPA), 33001 Oviedo, Spain

**Keywords:** theory of mind, reading the mind in the eyes test, psychopathy checklist-revised, comprehensive assessment of psychopathic personality

## Abstract

(1) Background: there is an ongoing debate about whether psychopathic traits increase or decrease cognitive empathy/Theory of Mind. (2) Methods: using a representative sample of 204 Spanish convicted inmates incarcerated at the Pereiro de Aguiar Penitentiary in Ourense, Spain, we investigated the relationship between two tools for the assessment of psychopathy, the Psychopathy Checklist-Revised (PCL-R) and the Comprehensive Assessment of Psychopathic Personality (CAPP), and the Reading the Mind in the Eyes Test (RMET), a well-known measure of cognitive empathy. (3) Results: The results showed no clear connection between the scores on the psychopathy assessment tools and RMET performance. This lack of association was stronger when the age variable was included in the multivariate analysis. (4) Conclusions: the results of this study failed to detect any clear link between psychopathy and cognitive empathy performance. Accordingly, our results indicate that psychopathy neither improves nor worsens cognitive empathy.

## 1. Introduction

Psychopathy is a developmental disorder defined by antisocial behavior paired with callousness, low empathy, and low interpersonal emotions [1]. Although not all psychopathic individuals are violent and not all violent offenders are psychopathic, these individuals are disproportionately involved in violence, and also in nonviolent criminal behavior [2]. The relationship between psychopathy and a shorter time to recidivism in criminal behavior has been demonstrated in both youth and adult incarcerated populations [3].

What is the main cause of these early aggressive and antisocial behaviors? The callous–unemotional component seems to be crucial [1] in creating an empathic dysfunction. This empathic dysfunction or emotional poverty has been clearly related to a reduction in specific forms of emotional empathy [4,5,6]. Emotional empathy involves affective responses to emotional displays of other people and to verbal descriptions of the emotional states of other individuals. Psychopathic individuals show a reduction in their capacity to respond to the fear, sadness, pain and happiness of others. They cannot use others’ emotions to regulate their own behavior, which makes them more predisposed to aggression and violence. This functional impairment is associated with reduced amygdala and ventromedial prefrontal cortex (vmPFC) responsiveness to distress cues [1,7].

What about the other component of empathy, i.e., cognitive empathy? This subtype of empathy applies to a set of reflective processes that include understanding, distinguishing another’s feelings from one’s own, and being able to integrate this information with social knowledge to adaptively guide interpersonal behavior [8]. Cognitive empathy shows many similarities to a subtype of Theory of the Mind (ToM), and the ability to attribute mental states to others. Affective ToM has been defined as the ability to make inferences about another´s emotional state. For many researchers, cognitive empathy and affective ToM are interchangeable [9]. Others point out that affective ToM only refers to an understanding of another´s state, while cognitive empathy requires active engagement in the another´s perspective [8]. However, all authors seem to agree that cognitive empathy and affective ToM share a neural network [9], that affective ToM is at least a prerequisite of cognitive empathy, and that cognitive empathy mediates the relationship between affective ToM and social functioning [10]. This subtype of empathy involves the representation of the intentions and thoughts of other individuals, also known as the theory of mind (ToM).

According to some researchers, being callous–unemotional is not related to a reduction in cognitive empathy and that is why psychopathic offenders show a preserved or even enhanced ability to recognize cues of emotional vulnerability in their victims [1,11]. However, other researchers disagree and point to data from studies in which cognitive empathy seems to be impaired in psychopathic individuals [11,12]. Why is this so? Is this related to the way the assessment was conducted? Or is this related to the other component of psychopathy, impairment in decision making, a general deficit in externalizing disorders, clearly related to impulsivity and antisocial behavior [1].

The Reading the Mind in the Eyes Test (RMET) is an advanced test that measures affective ToM through the interpretation of complex mental states only using the eye gaze [10]. Some authors suggest that the ability measured by the RMET might be more precisely described as emotion recognition rather than ToM [7]. If this is the case the RMET would be a tool for cognitive empathy evaluation.

Originally it was designed to assess affective ToM in autistic clinical samples [13]. It has also been used to assess affective ToM/cognitive empathy in psychopathic individuals. Mixed findings have made it hard to interpret the relationship between RMET and psychopathy. Some studies have found no difference at all in RMET performance between psychopathic and non-psychopathic individuals [14,15]. Other researchers have found significant relationships [11]. Different psychopathy assessment tools might explain the differences found.

The Psychopathy Checklist Revised (PCL-R) is the most well-known and used assessment tool in the field of psychopathy and has been declared the “measure of choice” [13]. It is a 20-item symptom construct expert rating scale, created for assessing psychopathy in forensic settings. Three-point scale ratings are given for lifetime presence, and the severity of each item is established through all available clinical and forensic data. Data are gathered through a semi-structured interview, file review, and collateral information. Research has shown that PCL-R, as a measure of psychopathy, has good psychometric properties [13]. However, doubts have emerged about the generalizability of its cutoff score, with clear differences between North American and European samples, and also about its predictive utility at the individual level, especially with regard to non-adult male serious offender samples, as were used for the PCL-R validation [16,17,18,19,20]. Concerns exist about the PCL-R relying too much on antisocial behavior to confirm who is a psychopath [19,21,22].

Researchers who do not view antisocial behavior as a key element of psychopathy have developed the Comprehensive Assessment of Psychopathic Personality (CAPP), a 33-item symptom construct expert rating tool, also designed for use in forensic settings [23,24]. Items are structured into six domains. Seven-point scale ratings are used for an assessment timescale, which usually ranges from 6 to 12 months. Data are collected through a semi-structured interview, file review, and collateral information. Research has shown that CAPP validity is high and generalizable across cultures and countries [25,26,27,28,29].

### Objectives

The present study combines PCL-R and CAPP assessment in an inmate sample in order to assess the relationship between RMET as a ToM assessment tool and two different dimensional psychopathy models. With this research strategy, we may be able to establish whether or not cognitive empathy is also impaired in psychopathy. Our hypothesis, following Blair´s model [1,4,5,6,30,31] was that affective ToM/cognitive empathy is preserved in psychopathic individuals.

## 2. Materials and Methods

### 2.1. Procedure

The protocol followed in the current study has been described in detail elsewhere [32,33,34,35]. Participants in the current study are the same as those in our previously published PCL-R and CAPP work [32,33,34,35]. The study was conducted at the Pereiro the Aguiar Prison, a low-medium security institution where all offenders from the Ourense region who receive aggregate sentences of 2 years or longer are incarcerated. The prison also houses inmates from other Spanish regions and prisons. Between April 2014 and April 2016, all convicted inmates were screened. Inclusion criteria were having served at least 6 months of their sentence at the Pereiro the Aguiar Prison and providing written informed consent. Exclusion criteria were not being a fluent Spanish speaker or having a serious neurological or psychiatric disorder.

### 2.2. Participants

In all, 330 inmates were screened. Of those, 126 (38.18%) did not meet the inclusion-exclusion criteria: 10 (7.93%) refused to participate and did not sign the written informed consent, 16 (12.69%) were not fluent Spanish speakers, 32 (25.39%) have been diagnosed with a serious neurological or psychiatric condition (15 with schizophrenia and related disorders, 10 with major affective disorders, and 7 with neurological cognitive impairment), and 68 (53.99%) had not served at least 6 months of their sentence at the prison. Thus, 204 (61.82%) inmates fulfilled the inclusion and exclusion criteria and were enrolled in the study. Then, of the 262 prisoners who fulfilled the inclusion criteria, 204 (77.82%) participated in the study, and only 10 (3.81%) refused to do so.

### 2.3. Instruments

The new data consist of a comparison of the RMET with the PCL-R and the CAPP.

A computerized version of the “Reading the Mind in the Eyes” Test—revised (Inquisit by Millisecond) was used to assess cognitive empathy (ToM). The test consists of 36 black-and-white images of people´s gazes. The images are presented one by one, together with four adjectives (one target adjective and three foils). The participants are instructed to select which of the four adjectives best describes what the person in the image is feeling (his or her mental state). The test is self-paced, and a glossary presenting a brief definition of each adjective was available if needed. The test is scored by adding up the number of correctly identified mental states. As in previous research [8], the gazes were also classified into three separate emotional valence categories (Positive, Neutral, and Negative). For comparison purposes, all the scores were divided by the number of stimuli in each category (Total = 36; Positive = 7; Neutral = 7; Negative = 17) [8].

All participants also completed the following protocol:-PCL-R: One of the researchers, GF, trained and experienced in the use of the PCL-R, interviewed all participants and coded the scores.-CAPP: One of the researchers, GF, trained and experienced in the use of the CAPP, interviewed all participants and coded the scores.-International Personality Disorder Examination (IPDE), DSM version: One of the researchers, GF, trained and experienced in the use of the IPDE, interviewed all participants and coded the scores.-Sociodemographic and forensic variables: The following variables were collected by researchers other than GF, blind to the PCL-R, CAPP and IPDE scores: Gender, age, nationality, number of years of education completed, marital status, total time in prison (months), drug/alcohol use (type, age of first use, principal route of administration), and type of official charges.

As previously indicated the protocol was approved by the Pontevedra–Vigo–Ourense Local Research Ethics Committee (2014/009) [21,22,23,24]. Every participant provided written informed consent. The study was conducted in accordance with the Declaration of Helsinki. No financial or other compensation was offered. Participants in the study were able to opt-out whenever they so wished. As there was no research treatment in the study, all inmates, whether participants or not, received the same treatments.

### 2.4. Data Analysis

R software (version 3.4.3) was used for all analyses (https://www.r-project.org/ (accessed on 1 april 2020)) [36]. Means and standard deviations and percentages were calculated for continuous and categorical variables, respectively. Group differences were found using the Mann–Whitney test for non-normally distributed variables. A correlation matrix was used to analyze the relationship between the PCL-R, the CAPP, and the RMET. Multiple regression analysis was used to explore the possible predictive power of the PCL-R and the CAPP on RMET performance. Both the PCL-R and the CAPP were independently analyzed as they measure the same underlying construct [11].

## 3. Results

The correlation analyses (Table 1 and Table 2) showed no significant relationship between the PCL-R and CAPP variables and the RMET variables, with one exception. There was a small significant correlation between PCL-R Facet 3 and RMET Total.

In the PCL-R multiple regression analyses, the following significant predictors of RMET performance were found (Table 3): for the RMET Total score, only the PCL-R Facet 3 score (t = 2.013, *p* = 0.045), for the RMET Neutral score, again the PCL-R Facet 3 score (t = 2.022, *p* = 0.044), and none for the REMT Negative and Positive scores.

In the CAPP multiple regression analyses, the following significant predictors of RMET performance were found (Table 4): for the RMET Total score, only the CAPP Cognitive score (t = −2.098, *p* = 0.037), for the RMET Neutral score, only the CAPP Behavioral score (t = 2.907, *p* = 0.004), none for the RMET Negative score, and finally, for the REMT Positive score, both the CAPP Dominance score (t = −1.982, *p* = 0.048) and the Self score (t = 2.053, *p* = 0.041).

Univariate analyses comparing the RMET variables with International Personality Disorder Examination (IPDE) diagnosis, drug/alcohol use, and type of criminal charges variables were also conducted. For the IPDE, the following significant relationships were found: RMET Total and IPDE Dependent (w = 39, *p* = 0.049), RMET Neutral and IPDE Borderline (w = 961.5, *p* = 0.035), RMET Neutral and IPDE Dependent (w = 28.5, *p* = 0.034), RMET Negative and IPDE Dependent (w = 28, *p* = 0.032). For drug/alcohol lifetime use, the following significant relationships were found: RMET Total and Methadone (w = 3824.5, *p* = 0.030), RMET Total and Cocaine (w = 3705.5, *p* = 0.002), RMET Total and Cannabis (w = 3974.5, *p*= 0.006), RMET Total and Hallucinogens (w = 1932, *p* = 0.039), RMET Neutral and Cocaine (w = 3970, *p* = 0.014), REMT Neutral and Cannabis (w = 4035, *p* = 0.009), RMET Neutral and Hallucinogens (w = 1796, *p* = 0.010), RMET Negative and Hallucinogens (w = 1935, *p* = 0.036), RMET Positive and Alcohol Use (w = 4136, *p* = 0.007), REMT Positive and Methadone (w = 3757, *p* = 0.014), RMET Positive and Cocaine (w = 4172.5, *p* = 0.045), and RMET Positive and Cannabis (w = 4324.5, *p* = 0.050). For the drug/alcohol use last month, no significant relationships were found. For the type of criminal charges, only the following significant relationship was found: RMET Negative and Drug Dealing (w = 4121.5, *p* = 0.042). The presence of the IPDE diagnosis (Dependent or Borderline), Drug Dealing, and lifetime use of the aforementioned drugs was always related to higher scores for the RMET variables, with the exception of Alcohol Use and RMET Positive.

It is well known that RMET performance is age dependent [37]. This relationship is also found in the present study, as can be seen in the following correlations between age and RMET variables: with RMET Total (−0.256, *p* < 0.001), with RMET Neutral (−0.251, *p* < 0.001), with RMET Negative (−0.166, *p* < 0.001), and finally with RMET Positive (−0.101, *p* = 0.150). Knowing this, age was introduced into the PCL-R and CAPP regression analyses (Table 5 and Table 6).

As can be seen in Table 5, only Age, and none of the PCL-R variables, were significant predictors of RMET performance: RMET Total (t = −3.093, *p* = 0.002), RMET Neutral (t = −2.806, *p* = 0.005) and RMET Negative (t = −2.475, *p* = 0.014). The same happens for the CAPP regression analysis: RMET Total (t = −3.177, *p* = 0.001), RMET Neutral (t = −2.319, *p* = 0.021), and RMET Negative (t = −2.677, *p* = 0.008); with the exception of Self in relation to RMET Positive (t = 1.990, *p* = 0.048).

Age also influences drug use, i.e., lifetime Cocaine (w = 6588.5, *p* = 0.001), Cannabis (w = 7124.5, *p* = 0.001) and Hallucinogens (w = 3234.5, *p* = 0.017) users are significantly younger than non-users.

## 4. Discussion

The current study researched the relationship between the RMET, a tool for assessing cognitive empathy, and two tools for the assessment of psychopathy, the PCL-R and the CAPP.

For the PCL-R, the correlation matrix revealed a positive trend only in the association between Facet 3 (lifestyle) and RMET Total (Table 1). The first regression analyses conducted revealed, once again, that only Facet 3 was positively associated with performance on the RMET (Total and Neutral) (Table 3). However, previous research has also found an association between PCL-R antisocial lifestyle scores and RMET scores [8], but to the contrary, it was a negative one. Contrary to previous research, in our study, interpersonal and affective PCL-R Facets (1 and 2) were not associated with any RMET scores. Furthermore, when age was introduced into the regression analyses (Table 5), even the positive association with lifestyle disappeared.

Something quite similar happened with the other psychopathy assessment tool, the CAPP. The correlation matrix (Table 2) showed no correlation between CAPP scores and RMET performance. The first regression analyses conducted (Table 4) revealed a negative association between CAPP Cognitive and RMET Total, and also a positive one between CAPP Behavioral and RMET Neutral. Once again, these findings are difficult to explain. Since CAPP Cognitive and Behavioral are domains closely related to one another, as previous CAPP research has shown [27,28,29,34], they should associate with RMET scores in the same way. For the CAPP Behavioral association with RMET performance, we found that, as with the PCL-R Facet 3, which measures a similar construct, it turned in the opposite direction when compared with previous research [11]. Again, when age was introduced into the regression analyses (Table 6), these associations were no longer significant. The same happened for the negative association between CAPP Dominance and RMET Positive (Table 4 and Table 6). Only a weak association between CAPP Self and RMET Positive remained significant when age was introduced into the regression analyses. Once again, this finding does not fit with results from previous research [11,38].

Taking all data together, our results indicate that psychopathy, assessed both with the PCL-R and the CAPP, does not influence RMET performance. Thus, psychopathy does not improve or worsen affective ToM/cognitive empathy.

The results from the present study fit with previous research that found no link between cognitive empathy/affective ToM performance and total psychopathy score [4,32]. Previous studies using the PCL-R also report a lack of a significant relationship between PCL-R Total score and RMET performance [8]. However, previous RMET studies have reported significant relationships when total scores were not considered, and subscores and dimensions were used both for the PCL-R and the RMET, or when other methods were used to assess psychopathy. In one of these studies [38], primary psychopathy was significantly negatively associated with RMET Total and Neutral performance and secondary psychopathy correlated negatively with RMET Total and Positive performance. However, this study used self-report measures to assess psychopathy in a small sample of female undergraduate students. In such samples, the prevalence of psychopathy is quite low, and it is difficult to generalize the results with other samples.

In another study [11], PCL-R Factor 1 (Facet 1 plus 2) was significantly positively associated with RMET Neutral performance, and Factor 2 (Facet 3 plus 4) was significantly negatively related with RMET Total, Neutral, and Negative performance. In that study, they also used a self-reported measure, and with that method, they found that interpersonal and affective traits (more related to primary psychopathy and PCL-R Factor 1) did not significantly predict performance on the RMET, while antisocial lifestyle (more related to secondary psychopathy and PCL-R Factor 2) significantly predicted a negative relationship to RMET Neutral and Negative performance. It is quite clear that there are important contradictions between studies and assessment tools when considering RMET performance. This second study used a sample closely similar to our study, i.e., prison inmates. The main difference is that they did not consider age when performing the regression analyses, even though the age of their participants ranged from 19 to 71 years (mean 33.47, SD 10.77 and in our study the mean age was 40.93, SD 11.18).

What about all the meta-analytical evidence indicating that emotion recognition deficits in psychopathy are present across emotions and modalities (facial and vocal) [39]? Well, that evidence does not contradict the results of the present study. This is because all the meta-analytical evidence is on the subject of recognizing basic emotions (emotional empathy, clearly impaired in psychopathic individuals) and is not related to RMET performance, a test that was designed for the interpretation of complex mental states only using the eye gaze (cognitive empathy or affective ToM) [40].

Nor did the present study find a clear relationship between affective ToM/cognitive empathy and the personality disorders assessed with the IPDE. The associations found with IPDE Dependent are not particularly strong (only two inmates matched this IPDE diagnosis). These results showed the weak relationship between CAPP Self and RMET Positive, and this is because CAPP Self is all about narcissism. Therefore, if this was a strong association, IPDE Narcissistic would also have had to be significantly associated with RMET Positive. The same applies to the type of criminal charges, i.e., no clear associations were found.

What about the baffling associations found between RMET performance and drug/alcohol lifetime use? At first glance, it could be concluded that lifetime abuse of Cannabis, Cocaine, and Hallucinogens improves affective ToM/cognitive empathy. This finding goes against all the research and literature on drug related brain damage. Fortunately, there is an easy and logical explanation for this finding. Once again, it is an age-related bias. Inmates who abuse these drugs are also younger and that is what actually improves their RMET performance [37].

The present study has some limitations that must be addressed. Although the sample used in the current study is bigger than the ones used in previous ones [11,38], its size can still be considered small, given the large number of variables analyzed. Another limitation is that the sample only includes inmates, which is inevitable when full versions of the PCL-R and CAPP are used. The generalizability of the findings is not guaranteed.

Our conclusions on the relationship between affective ToM/cognitive empathy are based on RMET performance. RMET has been criticized as its ecological validity is weakened by static images, the specificity of cues, and the forced-choice response format; however, the RMET validity is supported by strong associations with other social cognitive measures and strong test–retest reliability [4,15].

## 5. Conclusions

In conclusion, the results of the present study indicate that affective ToM/cognitive empathy measured with the RMET is not influenced by psychopathy, assessed with the PCL-R and the CAPP; nor by the presence of a personality disorder, assessed with the IPDE; nor by type of criminal charge or by drug/alcohol abuse. Future research using other tools that measure affective ToM/cognitive empathy is needed to confirm that psychopathy cannot be detected through an impairment in affective ToM/cognitive empathy.

Our results indicate that affective ToM/cognitive empathy is preserved in psychopathic individuals fulfilling Blair´s model.

## Figures and Tables

**Table 1 brainsci-12-01003-t001:** This table shows correlations between PCL-R (Total, Factor and Facet scores) and RMET (Total, Neutral, Negative and Positive scores).

	RMET Total	RMET Neutral	RMET Negative	RMET Positive
PCL-R Total	0.087 0.215	−0.012 0.863	−0.016 0.824	0.082 0.246
PCL-R Factor1	0.016 0.824	−0.102 0.147	−0.018 0.793	0.072 0.306
PCL-R Factor2	0.128 0.068	0.073 0.299	−0.009 0.9	0.068 0.336
PCL-R Facet1	0.037 0.602	−0.082 0.245	−0.001 0.985	0.031 0.659
PCL-R Facet2	−0.01 0.892	−0.101 0.152	−0.032 0.648	0.099 0.16
PCL-R Facet3	0.151 0.031	0.093 0.187	0.035 0.621	0.076 0.282
PCL-R Facet4	0.072 0.306	0.032 0.647	−0.063 0.37	0.043 0.538

*p* values are italicized. PCL-R: Psychopathy Checklist Revised; RMET: Reading the Mind in the Eyes Test.

**Table 2 brainsci-12-01003-t002:** This table shows correlations between CAPP (Total, Attachment, Behavioral, Cognitive, Dominance, Emotional and Self scores) and RMET (Total, Neutral, Negative and Positive scores).

	RMET Total	RMET Neutral	RMET Negative	RMET Positive
CAPP Attachment	0.019 0.784	−0.012 0.866	−0.067 0.342	0.108 0.124
CAPP Behavioral	0.04 0.569	0.117 0.097	−0.026 0.707	0.108 0.605
CAPP Cognitive	−0.062 0.381	−0.009 0.896	−0.072 0.307	0.019 0.782
CAPP Dominance	−0.051 0.468	−0.097 0.169	−0.012 0.866	0.018 0.803
CAPP Emotional	0.005 0.942	−0.009 0.898	−0.002 0.973	0.078 0.266
CAPP Self	0.033 0.644	−0.056 0.429	0.033 0.635	−0.022 0.108
CAPP Total	0.001 0.993	−0.013 0.857	−0.022 0.751	0.073 0.3

*p* values are italicized. CAPP: Comprehensive Assessment of Psychopathic Personality; RMET: Reading the Mind in the Eyes Test.

**Table 3 brainsci-12-01003-t003:** This table shows the PCL-R/RMET multiple regression analysis.

	Estimate	Std. Error	Beta	t-Value	Pr (>|t|)	Significance
RMET Total						
Facet 1	−0.016	0.189	−0.001	−0.085	0.932	
Facet 2	−0.142	0.188	−0.006	−0.754	0.451	
Facet 3	0.340	0.169	0.199	2.013	0.045	*
Facet 4	−0.068	0.204	−0.003	−0.333	0.739	
RMET Neutral						
Facet 1	−0.078	0.060	−0.107	−1.162	0.246	
Facet 2	−0.067	0.060	−0.009	−1.119	0.264	
Facet 3	0.109	0.054	0.199	2.022	0.044	*
Facet 4	−0.021	0.065	−0.003	−0.331	0.741	
RMET Negative						
Facet 1	0.004	0.049	−0.001	−0.089	0.930	
Facet 2	−0.016	0.048	−0.003	−0.335	0.738	
Facet 3	0.061	0.043	0.140	1.411	0.160	
Facet 4	−0.079	0.052	−0.141	−1.505	0.134	
RMET Positive						
Facet 1	−0.028	0.037	−0.007	−0.769	0.443	
Facet 2	0.050	0.037	0.001	1.352	0.178	
Facet 3	0.028	0.333	0.008	0.865	0.388	
Facet 4	−0.014	0.040	−0.003	−0.354	0.724	

CAPP: Comprehensive Assessment of Psychopathic Personality; RMET: Reading the Mind in the Eyes Test; Std. Error: Standard Error; * *p* < 0.05.

**Table 4 brainsci-12-01003-t004:** This table shows the CAPP/RMET multiple regression analysis.

	Estimate	Std. Error	Beta	t-Value	Pr (>|t|)	Significance
RMET Total						
Attachment	0.093	0.112	0.001	0.830	0.407	
Behavioral	0.112	0.089	0.172	1.613	0.108	
Cognitive	−0.264	0.126	−0.285	−2.098	0.037	*
Dominance	−0.163	0.093	−0.233	−1.750	0.817	.
Emotional	0.069	0.127	0.076	0.544	0.587	
Self	0.109	0.076	0.187	1.428	0.154	
RMET Neutral						
Attachment	0.015	0.036	0.050	0.423	0.672	
Behavioral	0.064	0.022	0.308	2.907	0.004	**
Cognitive	−0.046	0.040	−0.154	−1.146	0.253	
Dominance	0.049	0.029	−0.219	−1.660	0.098	.
Emotional	0.023	0.040	0.079	0.569	0.569	
Self	−0.009	0.024	−0.048	−0.370	0.711	
RMET Negative						
Attachment	−0.040	0.029	−0.168	−1.380	0.169	
Behavioral	0.005	0.018	0.030	0.285	0.776	
Cognitive	−0.045	0.032	−0.192	−1.400	0.163	
Dominance	−0.007	0.024	−0.039	−0.291	0.771	
Emotional	0.039	0.033	0.169	1.198	0.232	
Self	0.024	0.019	0.165	1.252	0.212	
RMET Positive						
Attachment	0.030	0.022	0.167	1.384	0.168	
Behavioral	0.001	0.013	0.010	0.100	0.920	
Cognitive	−0.024	0.024	−0.137	−1.007	0.315	
Dominance	−0.036	0.018	−0.026	−1.982	0.048	*
Emotional	0.007	0.024	0.042	0.300	0.764	
Self	0.030	0.014	0.269	2.053	0.041	*

CAPP: Comprehensive Assessment of Psychopathic Personality; RMET: Reading the Mind in the Eyes Test; Std. Error: Standard Error. * *p* < 0.05; ** *p* < 0.01.

**Table 5 brainsci-12-01003-t005:** This table shows the PCL-R/RMET multiple regression analysis including Age.

	Estimate	Std. Error	Beta	t-Value	Pr (>|t|)	Significance
RMET Total						
Facet 1	0.102	1.982	0.050	0.539	0.590	
Facet 2	−0.059	0.189	−0.028	−0.318	0.750	
Facet 3	0.147	0.177	0.086	0.834	0.405	
Facet 4	−0.121	0.200	−0.054	−0.606	0.542	
Age	−0.116	0.037	−0.241	−3.093	0.002	**
RMET Neutral						
Facet 1	−0.036	0.060	−0.054	−0.592	0.554	
Facet 2	−0.043	0.060	−0.063	−0.723	0.470	
Facet 3	0.053	0.057	0.097	0.937	0.349	
Facet 4	−0.037	0.064	−0.052	−0.578	0.564	
Age	−0.033	0.012	−0.219	−2.806	0.005	**
RMET Negative						
Facet 1	0.020	0.049	0.038	0.411	0.681	
Facet 2	0.001	0.048	0.001	0.019	0.984	
Facet 3	0.021	0.046	0.049	0.466	0.641	
Facet 4	−0.090	0.052	−0.160	−1.732	0.084	
Age	−0.023	0.001	−0.196	−2.475	0.014	*
RMET Positive						
Facet 1	−0.018	0.037	−0.045	−0.476	0.635	
Facet 2	0.057	0.037	0.139	1.539	0.125	
Facet 3	0.011	0.035	0.034	0.324	0.746	
Facet 4	−0.019	0.040	−0.044	−0.473	0.637	
Age	−0.010	0.007	−0.110	−1.383	0.168	

CAPP: Comprehensive Assessment of Psychopathic Personality; RMET: Reading the Mind in the Eyes Test; Std. Error: Standard Error; * *p* < 0.05; ** *p* < 0.01.

**Table 6 brainsci-12-01003-t006:** This table shows the CAPP/RMET multiple regression analysis including Age.

	Estimate	Std. Error	Beta	t-Value	Pr (>|t|)	Significance
RMET Total						
Attachment	0.080	0.110	0.086	0.729	0.466	
Behavioral	0.019	0.074	0.029	0.259	0.796	
Cognitive	−0.219	0.124	−0.237	−1.773	0.077	.
Dominance	−0.075	0.095	−0.107	−0.792	0.429	
Emotional	0.040	0.125	0.049	0.326	0.744	
Self	0.096	0.075	0.164	1.281	0.210	
Age	−0.118	0.037	−0.246	−3.177	0.001	**
RMET Neutral						
Attachment	0.012	0.035	0.040	0.340	0.734	
Behavioral	0.042	0.023	0.203	1.782	0.076	.
Cognitive	−0.035	0.040	−0.119	−0.889	0.375	
Dominance	−0.028	0.030	−0.127	−0.934	0.351	
Emotional	0.016	0.040	0.056	0.407	0.684	
Self	−0.012	0.024	−0.064	−0.503	0.615	
Age	−0.227	0.012	−0.180	−2.319	0.021	*
RMET Negative						
Attachment	−0.043	0.028	−0.180	−1.501	0.134	
Behavioral	−0.015	0.019	−0.091	−0.793	0.428	
Cognitive	−0.036	0.032	−0.151	−1.110	0.268	
Dominance	0.012	0.024	0.068	0.493	0.622	
Emotional	0.033	0.032	0.142	1.021	0.308	
Self	0.021	0.019	0.146	1.120	0.263	
Age	−0.026	0.009	−0.210	−2.677	0.008	**
RMET Positive						
Attachment	0.029	0.022	0.162	1.343	0.181	
Behavioral	−0.004	0.014	−0.039	−0.337	0.736	
Cognitive	−0.021	0.024	−0.120	−0.879	0.381	
Dominance	−0.030	0.019	−0.220	−1.584	0.115	
Emotional	0.005	0.025	0.031	0.221	0.825	
Self	0.029	0.014	0.261	1.990	0.048	*
Age	−0.008	0.007	−0.085	−1.084	0.280	

CAPP: Comprehensive Assessment of Psychopathic Personality; RMET: Reading the Mind in the Eyes Test; Std. Error: Standard Error; * *p* < 0.05; ** *p* < 0.01.

## Data Availability

Not applicable.

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
