# Peer review of "The Influence of Psychopathy on Incarcerated Inmates’ Cognitive Empathy"

_brainsci, 2022, doi:10.3390/brainsci12081003_

Round 1
Reviewer 1 Report
The researchers explore the relationship between psychopathic traits and cognitive empathy in a sample of inmates. Overall, the rationalization of the study should be conceptualized more clearly, considering the various constructs the researchers investigate, previous research, and the potential confound of ToM/cognitive empathy (or just empathy), especially given the tasks/tests used. The reason for the selection and conduction of statistical analyses is also not clear, and multiple pieces of information should be added about the sample, the analyses, and the results to obtain a better understanding of the findings. Overall, the results reported cannot be trusted until the issues described below are corrected.
Major
- The statement “The relationship between psychopathy and a shorter time to recidivism in criminal behavior has been demonstrated in both youth and adult incarcerated populations [3].” Can be expanded and clarified. Do the authors mean the relationship between higher psychopathic traits and shorter time to recidivism? Additional references to this relationship would reinforce the statement.
- The “callous unemotional component” requires further description. Is this component (construct?) different from cognitive empathy/ToM?
- In the abstract and throughout, the authors seem to use cognitive empathy and ToM as interchangeable concepts, but most researchers define empathy as a specific domain of ToM, or even a separate trait (e.g.., Quesque & Rossetti, 2020; Schaafsma et al., 2015). The authors should describe the differences among these constructs and what domain of social cognition they are studying specifically. Since they use the RMET, there will be a strong perceptual cognitive component.
- The authors should include a discussion considering criticisms of the RMET (Altschuler et al., 2021, Oakley et al., 2016) as a test of ToM as opposed to lower processing skills and discuss how this might have affected their findings.
- It would be helpful to have a more in-depth discussion of unemotional component as opposed to cognitive empathy in the literature. The authors make the point that cognitive empathy is not related to lack of emotionality and, in fact, could have a negative relationship. There should be a discussion of differences and similarities between the two constructs as well as a discussion of previous findings from research studies.
- In the statement Component of empathy, i.e., cognitive empathy, I don’t think empathy equals cognitive empathy. In fact, many researchers differentiate at least between affective and cognitive ToM (e.g., Abu-Akel & Shamay-Tsoory, 2011; Poletti et al., 2012)
- Authors should explain in what way(s) being unemotional is not related to reduction in cognitive empathy
- Why is this so? Is this related to the way assessment was conducted? Or is this related to the other component of psychopathy, impairment in decision making, a general deficit in externalizing disorders, clearly related to impulsivity and antisocial behavior[1].
o For this and other questions the authors formulate (e.g., line 45), they should instead address them by describing the research conducted to answer these questions or similar ones. For example, describing decision making impairment vs the components of psychopathy, vs externalizing disorders, vs antisocial behavior in the context of ToM. Further, the authors do not assess whether the relationship between psychopathy and empathy is caused/mediated by any of these variables so it is not clear why they are brought up.
- The authors do not introduce PCL-R in the introduction, unlike the other two tests they use. The full name of the tool should also be included at this point.
- As a main framework of the study, Blair ́s model should be introduced in the literature review and the reasoning behind the study should be delineated in terms of this framework.
- The reasoning for the study as described in lines 74-77 is not entirely clear from the introduction. It seems that what the authors are attempting to study has been done before according to their introduction. Is there a specific novel contribution in this study? How would the study contribute to clarifying some of the open questions in the literature?
- It would be helpful to report a power analysis for the final sample for the analysis that required the most power (multiple regression).
- Please report analyses and findings of normality tests that were conducted to assess nonnormality.
- The authors should explain why they conducted multiple regression in addition to Mann-Whitney U (nonparametric t-test). Was the MW-U to assess something different? What analyses specifically tested the hypothesis posed at the beginning by the researchers?
- Since the authors indicate that the sample was not normally distributed, what type of regression was used to test the data? Did the authors also use a different type of regression based on the distribution of the sample? Considering the non-normality of the data, a plot of the data distribution or some description of it should be included.
- Authors should indicate the type of correlation approach, total N, and the p-value cutoff in the tables.
- Did the authors have any partial or total missing data? How were outliers treated? Were any responses/participants eliminated based on outliers, etc.?
- The authors say that there were no significant correlations between RMET and the other tasks, however the correlation coefficient for CAPP TOTAL and RMET TOTAL is .993 and CAPP EMOTIONAL and RMET TOTAL is .942. This seems surprisingly high. Do the authors have an idea of why this value is so high? Has this been observed before? A correlation this high indicates that the two variables are almost entirely the same.
- In the multiple regression, authors should report B estimates, not t, in in-text descriptions. In addition, the final sample size entered in the regression, effect sizes for all effects, and p-value used in the regression should be indicated.
- It would be interesting to conduct a regression analysis examining the effect of CPP total and PCLR total on RMET total with an interaction term. While it is interesting to look at the subdimensions of the tools, the total scores are supposed to be indexes of ability, therefore the total scores are a way to answer the authors’ hypothesis that cognitive empathy is not impaired in psychopathic patients. By conducting a regression such that ^Y(RMET) ~ X1 (CPP) + X2 (PCLR) + X1*X2, you could identify if CPP and PCLR are independent predictors of RMET and whether the relationship is moderated.
- The directionality of the results should also be discussed, especially given that some subdomains have negative signs and others positive signs, and figures plotting the main regressions should be included.
- Why did the authors decide to conduct nonparametric t-tests for the IPDE and not a multiple regression? The authors haven’t discussed until now the purpose of the IPDE for their hypothesis. Given that they are testing subdomains of the scale, just like for CPP and PCLR, it would be more adequate to follow the same analyses as for the other variables. Also, please report if these analyses were conducted on the same N.
- Are the authors concerned about Type I error given the multiple comparisons of the MW-U’s for IPDE? Why not conducting an ANOVA (or Kruskal-Wallis for nonparametric) and then do posthoc comparisons?
- Authors should also report effect sizes for all Mann-Whitney tests, especially considering that most p-values are close to a .05 value even though they have a large sample size. Also, I believe the correct reported statistic is a U, not a W, even if in R the results are indicated with a W. Information of directionality of the results should also be included.
- There is no discussion about the results as they are reported which could help keep track of the findings.
- The authors don’t report descriptive statistics of the predictor variables and demographics. For example, what is the age range and mean that the authors include in the regressions in table 5? Also, could the authors elaborate on the age correlations and why they might matter in this age range? Age differences are found especially in children and older adults, so perhaps knowing the mean age of the sample would clarify this. Descriptive statistics would also be helpful to understand the degree of “psychopathy” in the sample. While all inmates participated in the study, it is likely that most of the inmates did not present psychopathic traits. Perhaps this is the reason why the finding presented different directions and mixed results?
- Overall, the results could be organized better, following subheadings probably, so that it is clear what type of analyses are conducted on what variables and why.
- I am not sure that it can be implied from the findings that cognitive empathy is not influenced by psychopathy, both because of the different significant predictors with different signs and the fact that one test only was used to assess cognitive empathy, especially one that relies so heavily in facial cues and lower-level processing. I think the authors should reframe the implications of their findings so that they reflect the limitations.
- Similarly, I am not sure that the explanation on meta-analysis evidence of emotion recognition is convincing given that the RMET is largely thought to be a task of emotion recognition (Quesque & Rossetti, 2020)
Minor
- The authors should add some information about potential bias from cases of non-violent psychopathic individuals that are undercounted because they do not express violent behaviors.
- Citation 1 in line 36 should be at the end of the sentence.
- Disagre disagree in line 50
- I recommend substituting callous – unemotional for just unemotional or lack of emotionality that carries a less negative connotation
- Explain exactly where the Ourense region is
- Pereiro the Aguiar Prison Pereiro de Aguiar Prison
- research treatment experimental treatment?
- Group differences were found Group differences were tested?
- The title of the first table is not in the right font/size
- REMT in line 162
Author Response
Dear reviewer,
Here you can find a new version of the manuscript, changes have been underlined. We have only adressed the changes that we have considered necessary, and we have ignored the rest.
Best

Reviewer 2 Report
This is an interesting manuscript. A number of considerations are set out below:
- A priori it is not necessary to make explicit the name of the section (introduction, method...) in an abstract.
- The introduction is interesting but a more in-depth analysis is needed. There are few references that justify the theoretical framework and it is necessary to expand.
- The section on methodology should be restructured so that the following sections appear clearly differentiated: Participants, Procedure, Instruments and Data Analysis.
- The section on Results is very complete and allows us to obtain much relevant information from the research.
- With respect to the discussion, more emphasis should be placed on future lines of research and their applicability.
- It should be made explicit whether the hypothesis is fulfilled according to Blair's model.
Thank you very much for your attention.
Author Response
Dear reviewer,
Here you can find a new version of the manuscript, changes have been underlined. We have only adressed the changes that we have considered necessary.
Best

Round 2
Reviewer 1 Report
The revised version presents a more complete picture of the background and terminology issues of the subject of TOM. I suggest revision of minor typos/spelling but otherwise the manuscript has been sufficiently improved.
Reviewer 2 Report
I recommend the manuscript for publication. Just as a last suggestion, I highlight that it would be very interesting to explicit that the hypothesis has been confirmed, almost with these words, so that the readers understand the last sentence on page 10 means concretely this point. In case the authors want to add this, it should be exposed on the section Discussion. Thank you very much for improving the effort and attention.